# Kinetics and Modeling of Counter-Current Leaching of Waste Random-Access Memory Chips in a Cu-NH_3_-SO_4_ System Utilizing Cu(II) as an Oxidizer

**DOI:** 10.3390/ma16186274

**Published:** 2023-09-19

**Authors:** Peijia Lin, Joshua Werner, Zulqarnain Ahmad Ali, Lucas Bertucci, Jack Groppo

**Affiliations:** Department of Mining Engineering, University of Kentucky, 310 Columbia Ave., Lexington, KY 40506, USA; peijia.lin@uky.edu (P.L.); zulqarnain.a.ali@uky.edu (Z.A.A.); lfbe222@g.uky.edu (L.B.); john.groppo@uky.edu (J.G.)

**Keywords:** circuit boards, recycling, copper leaching, ammoniacal solution, kinetic modeling, diffusion-controlled

## Abstract

The leaching of Cu in ammoniacal solutions has proven an efficient method to recover Cu from waste printed circuit boards (WPCBs) that has used by many researchers over the last two decades. This study investigates the feasibility of a counter-current leaching circuit that would be coupled with an electrowinning (EW) cell. To accomplish this objective, the paper is divided into three parts. In Part 1, a leaching kinetic framework is developed from a set of experiments that were designed and conducted using end-of-life waste RAM chips as feed sources and Cu(II)-ammoniacal solution as the lixiviant. Various processing parameters, such as particle size, stirring rates, initial Cu(II) concentrations, and temperatures, were evaluated for their effects on the Cu recovery and the leaching rate. It was found that the particle size and initial Cu(II) concentration were the two most important factors in Cu leaching. Using a 1.2 mm particle size diameter and 40 g/L of initial Cu(II) concentration, a maximum Cu recovery of 96% was achieved. The Zhuravlev changing-concentration model was selected to develop the empirically fitted kinetic coefficients. In Part 2, kinetic data were adapted into a leaching function suitable for continuously stirred tank reactors. This was achieved via using the coefficients from the Zhuravlev model and adapting them to the Jander constant concentration model for use in the counter-current circuit model. Part 3 details the development of a counter-current circuit model based on the relevant kinetic model, and the circuit performance was modeled to provide a tool that would allow the exploration of maximum copper recovery whilst minimizing the Cu(II) reporting to electrowinning. A 4-stage counter-current circuit was modeled incorporating a feed of 35 g/L of Cu(II), achieving a 4.12 g/L Cu(II) output with 93% copper recovery.

## 1. Introduction

Waste printed circuit boards (WPCBs), as a major constituent of electronic waste, contain approximately 30% metallics and 70% non-metallics by weight [1,2]. The abundance of Cu in WPCBs, varying from 10 to 30%, has made WPCBs a promising secondary resource [3,4]. The recycling of Cu from WPCBs has attracted increasing attention during the last two decades [5,6]. As a more sustainable process, ammonium salts ((NH_3_)_2_SO_4_, (NH_3_)_2_CO_3_ and NH_4_Cl) and ammonium hydroxide (NH_4_OH) have shown a satisfactory performance for Cu recovery from electronic wastes, as reported in the literature [7,8,9,10,11,12]. The merits of using ammoniacal systems to extract Cu from WPCBs are: (1) a higher selectivity towards Cu and other base metals, such as Zn, Ni, Co, etc.; (2) lower solubilities for common contaminants in WPCBs, such as Fe and Al; (3) a lower corrosivity from the equipment used in the process; and (4) fewer harmful byproducts and wastes generated, in comparison to acid systems.

To date, extensive research has been conducted to explore Cu leaching in ammoniacal solutions [3,7,13,14,15]. The concept of leaching Cu as Cu(I)-ammine under an oxygen-eliminated environment was also studied preliminarily by Oishi et al. [3,16,17,18]. In the subsequent electrowinning (EW), it was proposed that a significant amount of energy could be saved via depositing Cu from Cu(I)-ammine compounds [19]. The theoretical principles supporting the feasibility of this system were previously reported by Koyama et al. [14]. Accordingly, the higher oxidation–reduction potential (ORP) of Cu(NH_3_)_4_^2+^/Cu^0^ in comparison to the ORP of Cu(NH_3_)_2_^+^/Cu^0^ indicates the capability of Cu(NH_3_)_4_^2+^ to oxidize Cu^0^ as Cu(NH_3_)_2_^+^ in the presence of NH_4_^+^ [8,14]. The oxidation of Cu^0^ by Cu(NH_3_)_4_^2+^, and the formation of a stable Cu(NH_3_)_2_^+^ complex, are illustrated via Equations (1) and (2) [16]. In the proposed system, the leached Cu(NH_3_)_2_^+^ is then subject to the EW circuit where Cu^0^ is deposited on the cathode. The reported net potential (E_net_) of Cu(NH_3_)_4_^+^/Cu^0^ in an ammoniacal system is 0.20 V (as given in Equation (2)), which is much lower than that of Cu^2+^/Cu^0^ in an acidic Cu system (0.89 V) [14]. Electrochemically, the reduced net potential indicates a significant decrease in energy consumption when depositing Cu from Cu(NH_3_)_4_^+^, as opposed to traditional Cu EW from Cu^2+^.
Anodic reaction: Cu(NH_3_)_2_^+^ + 2NH_3_ = Cu(NH_3_)_4_^2+^ + e^−^(1)
Cathodic reaction: Cu(NH_3_)_2_^+^ + e^−^ = Cu + 2NH_3_(2)
Net reaction: Cu + Cu(NH_3_)_4_^2+^ = 2Cu(NH_3_)_2_^+^(3)

A simplified schematic utilizing the properties of this system is depicted in Figure 1 (adapted from references [7,16]). The primary consideration is the replenishment of Cu(I) in leaching, while depleting the Cu(II) as far as possible for the subsequent EW, via considering the direct EW of Cu from the lixiviant and utilizing Cu(II) as an oxidizer for continued leaching. In this arrangement, two significant parameters must be evaluated and controlled: (1) the Cu(II) concentration reporting to the cathode, and (2) the Cu(I) concentration. The first will largely dictate the current efficiency and kinetics, with significant impact from the second. 

From the literature, Koyama et al. (2006) considered the deposition of copper from Cu(I) systems [20]. They looked to determine the effect of the Cu(I) concentration on current efficiency over a range from 6.35 to 57.11 g/L (0.1 to ~0.9 mol/L), and found that the current efficiency was maximized in the range from 19.04 to 38.07 g/L (0.3 to 0.6 mol/L). They further investigated the effect of Cu(II) from 0 to 31.73 g/L (0 to 0.5 mol/L) vs. the current density, showing a decrease in current efficiency corresponding to an increase in Cu(II) concentration. The corresponding current densities as part of their experimentation reached the rather high level of 1500 A/m^2^.

In another work investigating the effects of various salts, Oishi et al. (2007a) studied several parameters of interest on the performance of EW on ammonia, sulfate, chloride, and nitrate salts and the effect of current density on efficiency and power consumption [16]. Of great interest is the near-linear relationship shown between Cu(II) concentration and decreased current efficiency. Although, in the chloride system, the concentration of Cu(II) should be roughly representative of either the sulfate or chloride system, the findings indicated that, in a 1 mol/L total copper system, the current efficiency ranged from nearly 100% to ~25% in a nearly linear fashion, corresponding to a concentration from ~3.17 to 34.09 g/L (~0.05 to 0.55 mol/L) of Cu(II).

Following work from this same group, Oishi et al. (2007b) focused on the deposit purity utilizing solvent extraction to purify the electrolyte, utilizing 44 g/L and 36 g/L for the ammonia sulfate and ammonia chloride systems, respectively [3]. The current density in this study was a more moderate 200 A/m^2^.

Lastly, from this group, Oishi et al. (2008) described an EW cell composed of a graphite felt anode arranged in a flow-through configuration to remove the oxidized Cu(II) [19]. The composition of this test was performed at 63.56 g/L (1 mol/L) Cu(I). What is intriguing is the 50.77 g/L (0.8 mol/L) Cu(I) achieved in the cell discharge returning to leaching, which is good for leaching. An alternative cell arrangement to achieve a similar result was disclosed by Werner (2021) in US Patent App. 17/797,687 [21]. Additionally, Sun et al. (2017) explored the electrochemistry of this system, providing diffusion coefficients as a function of the copper concentration in the range of 10–80 g/L [22].

To achieve maximum current efficiency and minimum power consumption in the EW stage of copper recovery, the maximum Cu(I) with a minimum amount of Cu(II) conditions is required. From the previous EW references, it is apparent that a 20–40 g/L range of Cu(I) and as low a concentration of Cu(II) as possible are ideal for the EW of ammoniacal copper in the lixiviants referenced. Because of the requirements to both minimize the Cu(II) concentration in the lixiviant post leaching, and to maximize recovery, a counter-current leaching arrangement is required. In the typical processing of solids such as E-waste, a typical mixer/settler arrangement is utilized to provide solid/liquid separations between stages. Owing to the need for anerobic conditions, combined mixer/settler units, such as those described by Werner (2023) in US Patent 11,596,912 and Werner (2021) in US Patent 11,207,614, may be of benefit [23,24].

To understand and develop a preliminary design of a leaching circuit, a predictive counter-current leaching model is necessary. Although counter-current leaching would appear to be a significantly important industrial process, with a wealth of information with regard to its use and adaptation to various uses, little literature has been found by the authors addressing this area. A few examples can be referenced by Klumpar et al. (1973) [25], and the work by Liu et al. (1983) [26], with some work being conducted in the gold recovery area of carbon-loading by Wadnerkar et al. (2015) [27].

Notably, Levenspiel (1998) stated that the general kinetics of a counter-current system can only be solved numerically, except for a special case of second-order kinetics [28]. With a Cu(I)-rich and Cu(II)-lean solution being the critical prerequisite to maximize the current efficiency in Cu EW, it is essential to develop a predictable model in order to simulate the behavior of Cu(I) and Cu(II) during leaching.

When considering how to model a counter-current leaching system, it becomes apparent that the foundation is the leaching kinetics. As the investigation of leaching kinetics was considered for this study, shrinking-core models (SCMs) were chosen as a starting point because they are often regarded as the most widely used kinetic model in many hydrometallurgical processes [29], and mechanistically describe leaching in the presence of an insoluble fraction. Within the existing literature, the SCMs under surface chemical reaction control, film diffusion control, and product layer diffusion control have been adopted and evaluated in different ammoniacal systems [30,31,32,33]. Most of these studies have focused on the ammoniacal leaching of Cu and base metals from primary ores. It has been found that the dissolution kinetics of target metals (Cu and Zu) in refractory sulfide ores were mainly chemical-reaction-controlled when elevated temperatures and pressure were used in leaching to break the mineral structure [31,34,35].

Only a few studies have focused on the ammoniacal leaching of Cu from secondary resources [7,36]. It was reported by Oluokun and Otunniyi (2020) that, when adopting H_2_O_2_ as a strong oxidizer, the dissolution kinetics of Cu in waste printed circuit board dust followed the chemical reaction and mixed reaction–diffusion-controlled mechanism [36]. Sun et al. (2015) characterized the leaching kinetics of Cu in information and communication technology (ICT) waste using ammoniacal carbonate medium under aeration [7]. An early kinetic stage, where O_2_ was slowly diffused from the bulk solution to the reaction interface, was revealed. The leaching kinetics of Cu in ammoniacal solution oxidized by the dissolved O_2_ were found to be controlled by mass transport.

However, the aforementioned studies only evaluated ammoniacal leaching under oxidative conditions utilizing oxygen. For a specific system under anaerobic conditions, where the oxidizer was strictly controlled as Cu(II) only, the development of a feasible kinetic model is needed. Further, the employment of complex waste PCBs may significantly impact the leaching kinetics. To achieve the goal of designing a counter-current leaching system for the electrowinning of Cu(I), this work will develop a three-part methodology to predict circuit performance. These are (1) a leaching kinetic framework from experimentation, (2) the adaptation of the experimental leaching data into a leaching function suitable for the reactors chosen, and (3) a counter-current leaching performance model. The specific objectives for each part include:

Part 1—Development of Cu leaching kinetics via experimentation:Evaluation of Cu ammoniacal leaching using end-of-life PCBs under anaerobic conditions;Design of experiments to determine leaching kinetics and kinetic model selection according to goodness of fit, considering the initial Cu(II) concentrations, particle sizes, stirring rates, and temperatures as the primary experimental factors.

Part 2—Adaptation of kinetics to a counter-current flow and stirred reactors:Extending the determined experimental model into one that can be utilized in a counter-current leaching model.

Part 3—Application of developed model of a CCL model:Development of a counter-current leaching model utilizing the kinetic leaching expression.

## 2. Materials and Methods

### 2.1. Theoretical Framework

#### 2.1.1. Part 1—Development of Cu Leaching Kinetics via Experimentation

For simplicity, the shredded Cu-containing chips may be viewed as spherical particles. In addition, considering the occurrence of Cu within board laminations and the solid–liquid interaction between the metallic Cu and Cu(II)-bearing solution, a shrinking core model, alongside a diffusion-controlled model, were taken into account to describe the reaction mechanism in such a system. In considering shrinking core models, five rate-limiting steps can be considered, consisting of: (1) the diffusion of the oxidizer Cu(II)-amine from the bulk solution to the interface of the non-reactive product layer; (2) the penetration of the Cu(II)-amine through the non-reactive layer, terminating at the reaction surface of Cu(0); (3) a chemical reaction at the interface where Cu(0) was oxidized to Cu(I); (4) the leached Cu(I) complexing with NH_3_ to form a soluble Cu(I)-ammine compound, and penetrating the product layer to reach the interface; and (5) the diffusion of the Cu(I)-amine complex from the interface to the bulk solution. The slowest among these five steps is considered to be the rate-controlling step. A schematic illustration of these five rate-limiting steps is provided in Figure 2.

Considering the Cu-NH_3_ leaching system, several commonly used kinetic models were evaluated for goodness of fit using the experimental data from different particle sizes and initial Cu(II) concentrations. The models considered included film diffusion, product layer diffusion, surface reaction, and semi-empirical models [28,33,37,38]. However, most of them were not suitable for this specific Cu-ammoniacal leaching system due to a poor data/model fit. In terms of suitability, only the Zhuravlev model was considered as having theoretical suitability owing to the proposed batch-leaching experimentation. In other words, Cu(II) would be consumed anaerobically without regeneration. This simplistic allowance of the changing concentration of the reactant, without reliance on numerical methods, was appealing. In nearly all shrinking-core models (SCMs), the reactants are considered constant. In this work, including variance in the oxidizer through the experiment, it is appropriate to use a non-constant concentration model. Zhuravlev et al. proposed a diffusion-controlled model assuming the concentration of one reactant; $C_{0}$ is not a constant, but a factor of the reactant activity, varying with (1−α) [38]. As such, the change in concentration $C_{0}$ is proportional to (1−α), and the instant concentration at time *t* is $C_{0}(1-\alpha )$.
(4)$$\frac{dx}{dt}=\frac{DV_{m}C_{0}(1-\alpha )}{x}=\frac{k_{z}(1-\alpha )}{x}$$
(5)$$k_{z}=DV_{m}C_{0}$$

The final expression of the Zhuravlev changing concentration model can be given as:(6)$${\left({\left(1-\alpha \right)}^{-\frac{1}{3}}-1\right)}^{2}={k_{z}}^{\prime }t$$
where ${k_{z}}^{\prime }=\frac{2k_{z}}{{r_{0}}^{2}}=\frac{2DV_{m}C_{0}}{{r_{0}}^{2}}$; *r*_0_ is the original radius of the spherical particle A and *α* is the reacted fraction of Cu; *D* is the diffusion coefficient, m^2^/s; *V_m_* is the volume of product formed from 1 mole of the slowest penetrating component; *C*_0_ is the initial concentration of the reactant, mol/L, and the change in concentration is based on $C_{0}\times \left(1-\alpha \right)$ [38].

This model is of particular interest because of its approximation of the changing concentration of the reactant that diffuses towards the unreacted core [38]. Because of the high initial input and the continuous change in Cu concentration in the leaching system, the varying Cu concentration was believed to have the most significant effect on the leaching rate.

#### 2.1.2. Part 2—Adaptation of Kinetics to Counter-Current Flow and Stirred Reactors

The second part of the methodology aimed to adapt the changing concentration model coefficients to those suitable for constant concentration models. This is because the fundamental assumption of a continuously stirring tank reactor (CSTR) most often utilized in leaching assumes that the concentration in each individual leaching tank is constant. Therefore, a suitable means must be devised to adapt the results from a changing concentration kinetic model to those of a constant concentration model. These coefficients are shown explicitly in Equation (7).
(7)$${k_{z}}^{\prime }=\frac{2DV_{m}C_{0}}{{r_{0}}^{2}}$$

Correspondingly, a model for leaching under constant concentration was developed via adopting the interchangeable *k* value from the Zhuravlev changing-concentration model into the Jander constant-concentration model [39], which was chosen for the similarities in coefficients:(8)$${\left({1-\left(1-\alpha \right)}^{\frac{1}{3}}\right)}^{2}=\frac{2DV_{m}C_{0}}{{r_{0}}^{2}}t={k_{j}}^{\prime }t$$

In this manner, the coefficients solved in the Zhuravlev changing concentration model may be adapted to those in the Jander constant concentration model, for the purpose of demonstration, through assuming that ${k_{z}}^{\prime }={k_{j}}^{\prime }$. To further expand the utility of the model, these coefficients are modified via the consideration of the Arrhenius Equation (9) in the following manner.
(9)$$K=Ae^{-\frac{E_{a}}{RT}}$$
(10)$${k_{z}}^{\prime }={k_{j}}^{\prime }=k^{\prime }=b\frac{e^{-\frac{E_{a}}{RT}}C_{0}}{{r_{0}}^{2}}$$
or
(11)$$k^{\prime }=b^{\prime }C_{0}$$
where $C_{0}$ may be either the initial concentration in a changing concentration model or the constant concentration in the Jander model.

#### 2.1.3. Part 3—Application of Developed Model of a CCL Model

Having established the interchangeability of these coefficients, we may now consider their use in a counter-current leaching model. Assuming an integer of *i* corresponding to the reactor number with reactor 1 being where the unleached solids enter and the utilized lixiviant exits, the concentrations of oxidants (Cu(II)) may be defined as C_1_, C_2_, C_3_, …, C_n−1_, C_n_, where the integer n corresponds to the last reactor where the leached solid exits and the fresh lixiviant enters. Therefore, leaching commences at *t*_1_ = 0 and α = 0, and the reaction will continue until the particle leaves the reactor. When particles are transported to the next reactor, the α of the particle will remain the same between the two reactors but the concentration will change between them. This will introduce the particle into a new kinetic regime.

Because α is equivalent for the time when a particle leaves one reactor and when it enters another, the following equation is true, but must be considered in the context of *t_n−1,leaving_ ≠ t_n,entering_* because C_n−1_
*≠* C_n_.
(12)$$t_{i,\;entering}=\frac{{\left({1-\left(1-\alpha \right)}^{\frac{1}{3}}\right)}^{2}}{b^{\prime }C_{i}}$$

Because of the difference in the effective starting time to match α on tank transfers, an expression for α with regards to the elapsed residence time must be considered, as follows, when solving Equation (12) for α.
(13)$$\alpha =3{\left(k^{\prime }t_{i,\;leaving}\right)}^{\frac{1}{2}}-3(k^{\prime }t_{i,leaving})+{{(k}^{\prime }t_{i,leaving})}^{\frac{3}{2}}$$

With these two equations and the developed framework, the implementation may now be considered.

### 2.2. Waste RAM Chips

The waste RAM chips used in this study were obtained from the University of Kentucky Recycling Service. These chips were selected as the feed materials due to the high Cu and Au content in a relatively homogeneous form. The reduction in the size of the waste chips was achieved using a knife mill (Retch SM 300), with different sizes of interchangeable screens. The particle size was reduced to a top size of −3.4, −2.0, and −1.2 mm via the shredder over multiple passes. For further chemical assaying purposes, an analytical mill (Cole Parmer Analytical Mill 4301-00, Cole-Parmer Instrument Company Ltd., St. Neots, UK) was used to further pulverize the shredded chips to −600 μm.

### 2.3. Leaching Experiments Pertaining to Part 1

A leaching apparatus equipped with a heating and stirring mantle, a three-neck round-bottom vessel, and a pH/Eh/temperature probe were utilized for Cu ammoniacal leaching. The reaction vessel was connected to an air-tight sampling port, a gas-purging port, and a condenser with an outlet to a gas scrubber. Syringes were attached to the air-tight sampling port to take liquid samples during leaching while preventing the oxidation of the Cu(I) in the liquid due to contact with the air. The NH_3_ gas scrubber was filled with a known amount of H_2_SO_4_, to capture the evaporated NH_3_, and the amount of NH_3_ evaporation was determined from the pH shift of the prefilled H_2_SO_4_ solution. The solution Eh, pH, and temperature during the leaching process were monitored using a multi-functional pH/ORP/ATC (Automatic Temperature Compensation) probe (Mettler-Toledo InPro 3100i, Hongkong, China).

The leaching experiments were carried out using shredded waste chips in ammoniacal solution, with a solid/liquid ratio of 50 g/L, under atmospheric pressure for 8 h. According to the existing literature studying Cu ammoniacal leaching, lixiviant compositing (NH_4_)_2_SO_4_ ranging from 0.3 to 2 M, and NH_4_OH ranging from 4 to 6 M was sufficient to achieve a satisfying Cu recovery of over 90% [7,14,15,40]. In this work, an ammoniacal solution consisting of 1 M (NH_4_)_2_SO_4_ and 4 M NH_4_OH was employed. An amount of 1 M of (NH_4_)_2_SO_4_ was chosen to supply sufficient SO_4_^2-^ as the anions stabilize the leached Cu-amine species, and 4 M NH_4_OH was chosen to provide excessive NH_3_ ions in the solution as a complexing agent, in addition to conditioning the pH at a range from 9 to 11. In the ammoniacal lixiviant, the initial Cu(II) was made up to various concentrations via dissolving CuSO_4_·5H_2_O to serve as the oxidizer in facilitating Cu recovery. All the experiments were run under Ar covering gas to expel existing oxygen in the system, and to control the initial concentration of Cu(II) as the only oxidizer. The amounts of initial Cu(II) added at the beginning of leaching were 10, 20, 30, and 40 g/L. Other experimental parameters include the particle size (1.2, 2.0, and 3.4 mm), stirring rate (450, 600, and 750 rpm), and temperature (18, 25, 35, 45, 55, and 65 °C). In addition, the influences of the temperature and Cu concentration on the rate of NH_3_ evaporation were evaluated. The conditions of the bench-scale Cu ammoniacal leaching experiment are summarized in Table 1.

Samples were taken at 0, 15, 30, 60, 120, 240, and 480 min of leaching time. All the samples were filtered immediately to stop any further leaching and were prepared for ICP analysis to determine the elemental concentrations. The solid residues after leaching were rinsed and dried in an oven at 65 °C prior to a chemical digestion procedure to determine the metal contents remaining in the solid phase. De-ionized water was used in all processes. All chemicals used in the assay, sample preparation, and leaching experiments were reagent grade.

### 2.4. Chemical Assay and Analytical Methods

Due to the formation of precipitates in the transfer of the liquid samples from an alkaline condition (in leaching test) to an acidic condition (in the sample matrix required for ICP), a sample preparation procedure was developed to stabilize the species in liquid samples. Liquid samples (1 mL) were first acidified using 1 mL of concentrated HNO_3_ to stabilize the metal species as nitrates. The acidified sample was then oxidized using 1 mL of concentrated H_2_O_2_ to stabilize the Cu ions as Cu(II)-nitrate compound. Lastly, the stabilized liquid samples were topped off to 10 mL total volume (10× dilution) with de-ionized water.

Solid samples consisting of feed RAM chips and leaching residues were subject to a chemical assay procedure whereby the solids were pulverized with an analytical mill, roasted in a muffle furnace, and digested in HF and aqua regia on a hot block. The elemental concentrations in the digested solution were analyzed via ICP-OES (inductively coupled plasma optical emission spectrometry) using a Thermo Scientific iCAP 6500 Duo, Waltham, MA, USA, dual-view ICP-OES. The concentration of total Cu was determined via ICP-OES. The concentration of Cu(II) in the solution was measured via UV/Vis spectrophotometry, at a wavelength of 630 nm. A sampling port with syringes attached was used to prevent contact with the air and the oxidation of Cu(I) outside of the leaching vessel. The concentration of Cu(I) was calculated via the subtraction of the Cu(II) amount from the total Cu.

Because of the heterogeneity of the feed source, the Cu recovery was calculated via taking into account the Cu content in both the liquid phase and solid residues, as expressed by Equation (14):(14)$$Cu\;Recovery\;\left(\%\right)=\frac{{\left[Cu\right]}_{solution}\times V_{solution}}{{\left[Cu\right]}_{solution}\times V_{solution}+{\left[Cu\right]}_{residue}\times m_{residue}}$$
where [Cu]_solution_ is the concentration of the total ionic Cu leached in the solution (including Cu(II) and Cu(I)), in mg/L; V_solution_ is the volume of solution, in liters; [Cu]_residue_ is the remaining Cu concentration in the leaching residues, in mg/kg; and m_residue_ is the mass of the leaching residues, in kilograms.

## 3. Results and Discussions

### 3.1. Element Composition in Waste Chips

The obtained waste chips were shredded to −2 mm and used as feed material for Cu ammoniacal leaching. The shredded chips were further pulverized to −20 mesh and homogenized for chemical assay. The metal contents in the waste chips, determined via assay, are shown in Table 2. Copper was the highest by mass metal, accounting for 309,691 ppm (~30.97%wt.). There was also 15,256 ppm (~1.53%wt.) of Ni and 693 ppm of Au found in the RAM chips. The contents of the main containments, Fe, Al, and Pb, were 18,066 ppm (~1.81%wt.), 8629 ppm, and 2584 ppm, respectively. The chips also contained 1224 ppm of Co and 6448 ppm of Sn.

### 3.2. Effect of Leaching Parameters (Stirring Rate, Particle Size, Cu(II) Concentration, and Temperature)

Leaching experiments using shredded RAM chips were carried out to investigate the parameters of significance. A fixed amount of 25 g feed RAM chips, 500 mL lixiviant composed of 1 M (NH_3_)_2_SO_4_ and 4 M NH_4_OH, and consistent argon cover gas at 0.1 L/min were adopted in all the leaching tests. The effect of the stirring rate, at 450, 600, and 750 rpm, on the Cu recovery is shown in Figure 3. As these three curves from the variation in the stirring rate appear nearly identical, it seems that this variable, in these ranges, has little to no noticeable effect on Cu leaching. The average Cu recovery was 86% at these stirring speeds.

To study the effect of liberation and the impact on the leaching efficiency, waste chips were shredded to a maximum size of 3.4, 2.0, and 1.2 mm, respectively. The effect of the particle size is shown in Figure 4. As indicated, Cu recovery increased from 77% to 86%, and 93%, as the particle size decreased from 3.4 to 2.0, and 1.2 mm, respectively. As the particle size decreased, the surface area of the chips exposed to the lixiviant increased, causing a faster dissolution of Cu(0) into the solution. Although the highest Cu recovery was achieved with the 1.2 mm particle size, this was not selected for additional testing in the experimental series due to the high energy consumption related to mechanical shredding and the high surface tension (increased hydrophobicity), which caused difficulty in wetting the particles in leaching.

In an anerobic environment, the initial input of oxidizer (Cu(II)) was believed to have the most significant effect on the Cu leaching for both the recovery % and leaching rate. Based on the estimation of the amount of Cu(0) existing in feed (equivalent to 15.5 g/L at 100% recovery), the initial Cu(II) concentrations were chosen as 10, 20, 30, and 40 g/L. The results shown in Figure 5 demonstrate that, in the range from 10 to 30 g/L, Cu(II) plays an important role during leaching. Furthermore, increasing the initial Cu(II) concentration from 30 to 40 g/L slightly enhanced the Cu recovery from 84% to 86%. As the Cu(II) concentration increased, the rate of reaction also increased, which is indicated by the steepness at the beginning of the leaching curves (within the first 2 h). To maintain the optimum performance of Cu(II) oxidation, 40 g/L as the initial concentration was used in the subsequent tests.

The effect of temperature was also studied. Due to the high vapor pressure of NH_3_, the evaporation of NH_3_ from a solution (NH_4_OH) to gaseous phase (NH_3_) is increased at a higher temperature. Therefore, the temperature range selected was from 18 °C (natural temperature of lixiviant solution) to 65 °C. As revealed in Figure 6, elevated temperatures did not significantly affect Cu recovery or the leaching rate. The temperature influence can be divided into two ranges: a lower temperature zone from 18 to 35 °C, and a higher temperature zone from 45 to 65 °C. At lower temperatures, the average Cu recovery was 88%, while at higher temperatures, an average recovery of 93% was achieved.

### 3.3. Kinetic Modeling Fitting Corresponding to Part 1

To determine the goodness of fit, the Cu recoveries in % obtained from the batch leaching experiments, with regard to different particle sizes and changing Cu(II) concentrations, were converted to the reacted fraction, α (which ranges from 0 to 1, with 0 being unreacted, and 1 being fully reacted). The left term containing α, in each model, was plotted versus the reaction time. The linear regressions from the kinetic model are shown in Figure 7, Figure 8 and Figure 9, respectively. The slopes of each linear line, k, are the rate constants varied by the considered variables.

The regression of the Zhuravlev model with the changing C_0_ bulk concentration is shown in Figure 7 and Figure 8 and corresponding R^2^ shown in Table 3 and Table 4. The result indicates a good fit, with R^2^ above 0.99 for most settings in both sizes and initial concentrations. This appears to validate the assumptions of the Zhuravlev model and confirm its suitability.

The relationship between the reaction rate and temperature can be described via the Arrhenius equation, and the activation energy (E_a_) was evaluated in the context of the Zhuravlev changing-concentration model (Equation (9) [41]. To analyze the effect of the temperature, a plot of time vs. ${\left({\left(1-\alpha \right)}^{-\frac{1}{3}}-1\right)}^{2}$ was performed, with each series corresponding to a temperature (see Figure 9a). The regressed slopes, which corresponded to the rate constant ($k_{temp}^{\prime }$) from Figure 9a, were then plotted to determine the activation energy (E_a_), which is calculated via taking the natural logarithm of the rate constant (lnk_T_/h^−1^) versus the inverse of the temperature (T^−1^/K^−1^), as shown in Figure 9b. The value of the resulted slope can be depicted as −E_a_/R, where R is the gas constant of 8.314 J·K^−1^·mol^−1^. From this regression, the activation energy (E_a_) in the Cu-NH_3_ leaching system was determined to be 5.37 kcal/mol (or 22.49 kJ/mol), which suggests that this leaching reaction is primarily controlled by mass transport.

To further determine the dependency of the reaction order regarding various particle sizes and initial Cu(II) concentrations, within the studied range, the logarithm of apparent rate constants obtained from Figure 7 and Figure 8 was taken and plotted vs. $\ln\left[\left(\frac{1}{R^{2}})/(mm\right)\right]$ and $\ln\left[c({Cu}^{2+})/(mol/L)\right]$, as shown in Figure 10a,b, respectively. The estimated reaction order under different sizes ($\frac{1}{R^{2}}$) was 1.0089, while the reaction order under different Cu(II) concentrations ($c({Cu}^{2+})$) was 1.8885. This apparent reaction order will be discussed later, as a diffusion model is selected.

With the general goodness of fit established in Figure 9 and Figure 10, the development of a fitted kinetic model may be considered. These figures suggest that reasonable fitting may be expected. Recalling Equations (6) and (10), the proposed model is:(15)$$\left({\left({\left(1-\alpha \right)}^{-\frac{1}{3}}-1\right)}^{2}\right)=13.91\times \frac{1}{{r_{0}}^{2}}\times C_{0}\times exp(-\frac{2704.5}{T})\times t$$
where $C_{0}$ is the initial Cu concentration in the unit g/L; *r*_0_ is the particle size radius of feed materials in the unit mm; and T is the reaction temperature in K.

The above equation corresponds to the unit of g/L, K, and mm for inputs. If mol/L is used for concentration, the coefficient becomes 883.9. The activation energy simplification from Figure 9 should be noted. The coefficient was determined through comparing the experimental results with those predicted by the model and minimizing a sum of squares error. The resulting fit can be seen in Figure 11, which shows an R^2^ of 0.985 (using Equation (15)).

This is interesting, considering that the results in Figure 10 show the Cu(II) concentration influencing the reaction rate with an order of 1.8885. This is curious, as the model presented shows only a linear dependency on concentration. To evaluate this effect further, Equation (15) was modified with the fitted concentration dependency and tested with goodness of fit, which produced a slightly lower R^2^ of 0.983, seeming to suggest the correctness of the proposed model. For the sake of completeness, the alternate model in mol/L is presented:(16)$$\left({\left({\left(1-\alpha \right)}^{-\frac{1}{3}}-1\right)}^{2}\right)=1576.8\times {C_{0}}^{1.8885}\times {\left(\frac{1}{{r_{0}}^{2}}\right)}^{1.0089}\times \exp\left(-\frac{2704.5}{T}\right)\times t$$
where $C_{0}$ is the initial Cu concentration in the unit mol/L; *r*_0_ is the particle size radius of feed materials in the unit mm; and T is the reaction temperature in K.

Although they are not focused on E-wastes, similar evaluation methods are provided for reference, utilizing similar kinetic-fit approaches for Cu leaching from mineral matrices (such as chalcopyrite, bornite–chalcopyrite, arsenopyrite, etc.) in acidic environments [42,43,44]. Hidalgo’s work is particularly insightful, comparing more traditional “averaging” approaches to a more discrete analysis to identify distinct reaction mechanisms.

In a direct comparison considering Equation (15) versus Equation (16), the R^2^ of both model expressions showed an extremely similar goodness of fit (with R^2^ of 0.985 and 0.983). Because of the similarity of both models in terms of the feasibility of predicting the α, the original derivation of the Zhuravlev model with the first-order expression of $C_{0}$ and *r*_0_ (Equation (15)) was chosen for the condition of the CSTR CCL circuit. A more detailed evaluation and comparison of the two methodologies and their applications in batch leaching and CCL circuits is elaborated in the work by Lin [45].

### 3.4. Modeling Leaching in a Counter-Current Circuit as It Relates to Part 3

With the kinetic model (Equation (15)) developed, and the transposition methodology defined, (Equations (8), (12) and (13)), an Excel-based leaching model was developed to demonstrate the concept. The developed model is shown in Figure 12, which shows the model developed utilizing the sum of squares minimization of the concentration to provide a solution.

Through applying the following input parameters (as listed in Table 5) for a counter-current leaching circuit using continuous stirred-tank reactors (CSTRs), the cumulative reacted fraction α was obtained and is listed in Table 6. The estimated reaction fraction α listed in Table 6, showed a depleting trending for Cu(II) from Tank 4 (most concentrated) to Tank 1 (most depleted). The cumulative α at the end of leaching approached 0.93 (93% Cu recovery). 

A significant change to be noted in the inputs is the rate coefficient, now held as a variable which differs according to the concentration in each tank. The radius of particles is 1 mm (top size diameter of 2 mm), and the reaction takes place under ambient temperature. The starting lixiviant, composed of 35 g/L Cu(II) and 5 g/L Cu(I), enters the counter-current leaching circuit in Tank 4, at a flow rate of 500 L/min. The pulp density is 10.58% (solid/liquid ratio of 0.1058). The solid phase, containing 30%wt. of Cu(0), flows through the circuit from Tank 1 at a mass flow rate of 3.33 t/h for the total solid, and 1 t/h for the Cu(0), respectively. The rate coefficient, varying under different leaching conditions, was calculated to be 0.001363, according to Equation (15).

The estimated α was then programmed in a mass-balanced flowsheet created in Excel, as shown in Figure 13. As presented, the fresh lixiviant composing about 35 g/L Cu(II) and 5 g/L Cu(I) enters the leaching circuit from Tank 4. The waste PCBs, containing 30% wt. of Cu(0), are fed into the circuit from Tank 1, with 0 fraction reacted. The reacted solids in Tank 1 are then transferred to Tank 2, where the Cu(II) concentration is higher. By the time the materials reach Tank 4, the remaining Cu(0) is near nil. As it hits the highest Cu(II) concentration of 35 g/L, the residual Cu(0) is readily extracted in such a concentrated solution, optimizing the total recovery from 89% in the batch leaching circuit to 93% in the CCL circuit. At the end, the pregnant leaching solution, consisting of approximately 4.12 g/L of Cu(II) and 35.88 g/L of Cu(I), enters the EW cell, where the current efficiency benefits from the Cu(II)-depleted and Cu(I)-enriched solution.

The resulting Cu concentration (g/L) after each stage of leaching, estimated via the model, is shown in Figure 13. As shown, when started with 35 g/L Cu(II) and 5 g/L Cu(I) from the EW return, the stabilized final Cu(I) concentration when reaching a steady state in the continuously stirring tank reactor (CSTR) was about 35.88 g/L. Likewise, after each stage of leaching, the final concentration of Cu(II) gradually decreased, eventually to about 4.12 g/L, when leaving the leaching circuit for EW. As indicated by this trend, there should be a change in the rate constant k, corresponding to the change in Cu(II) concentration. In other words, the predicted model is changed from a higher concentration to a lower concentration when transferring from tank to tank.

In the counter-current leaching circuit, starting from Tank 1, the solid is the most intact and the lixiviant is the most depleted in the oxidizer, and the reverse is true for Tank 4. In this case, the predicted value for the reacted fraction in Tank 1 undergoes the constant concentration model, where the initial reactant concentration is 4.12 g/L, as shown in Figure 14 (blue curve). It should be noted that the retention time in Tank 1 is 3.2 h, which is sufficient to achieve the same recovery as in the next leaching tank. Subsequently, in Tank 2, the starting concentration fell (the orange curve), under an in-tank concentration of 15.80 g/L. Similarly, as Tank 4 is the beginner tank with the highest concentration of Cu(II) and the last tank with most diluted Cu(0) in a solid state, the predicted model therefore fits in the later retention time of the yellow curve. In fact, the operation manner in the counter-current leaching circuit resulted in a “zig-zag” shape of the leaching curve, where the α corresponds to two different times when the solid changes tanks that have two different concentrations, as shown in Figure 15.

## 4. Conclusions

In this study, Cu recovery from real waste chips, using ammoniacal solution with Cu(II) as the oxidizer, was investigated. Batch-leaching experiments were designed and implemented to explore the key processing parameters. The highest-observed Cu recovery of 96% occurred under the conditions of a 50 g/L solid/liquid ratio, 1.2 mm particle size, 750 rpm stirring rate, and 8 h residence time at room temperature (18 °C), using 1 M ammonium sulfate and 4 M ammonium hydroxide as the lixiviant, and 40 g/L initial Cu(II) concentration as the oxidizer.

To provide an in-depth understanding of the leaching kinetics and the Cu oxidation state, leaching mechanisms were evaluated, and kinetic modeling was executed. The resulting reaction rate under various parameters indicates that the particle size and initial Cu(II) concentration both showed the most significant impacts on kinetics. Indicated via kinetic modeling using experimental data, the Zhuravlev changing concentration model showed the goodness of fit. In considering the actual leaching mechanism, it appears that the change in concentration during leaching has the most significant effect on the leaching rate.

The activation energy was calculated to be 5.374 kcal/mol (22.485 kJ/mol) using the Arrhenius equation, which further confirmed that this reaction is dominantly controlled by mass transport. The general expression of Cu leaching in the ammoniacal system was established within the studied range of variables. The initial Cu(II) concentration was proven to be the most important factor influencing the Cu leaching rate, with the particle size second. Lastly, the comparison between the datasets predicted by the model and the data obtained from the experiment further supported the feasibility of the proposed model expression. Lastly, the Excel model shows promise in being able to explore the design parameters needed to maximize copper recovery and minimize the Cu(II) leaving the circuit.

## Figures and Tables

**Figure 1 materials-16-06274-f001:**
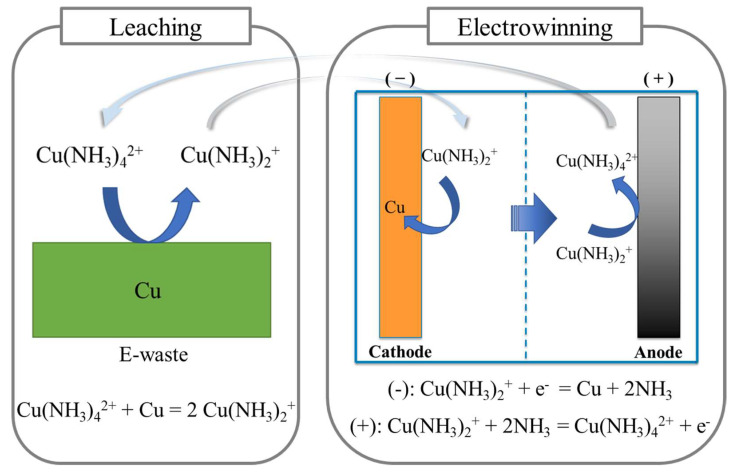
Schematic illustration of the coupled leaching-EW circuit in the Cu(II)-NH_3_-SO_4_ system. Adapted from references [7,16].

**Figure 2 materials-16-06274-f002:**
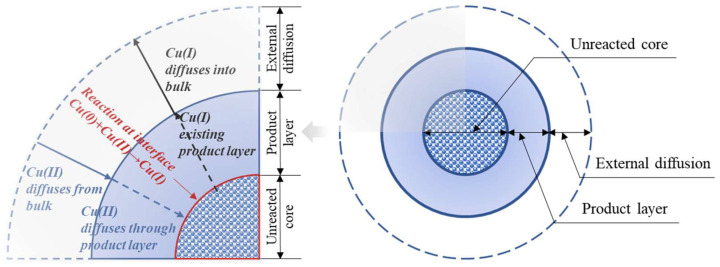
Schematic illustration of the mechanism and kinetic considerations in a Cu-ammoniacal leaching system. Adapted from reference [28].

**Figure 3 materials-16-06274-f003:**
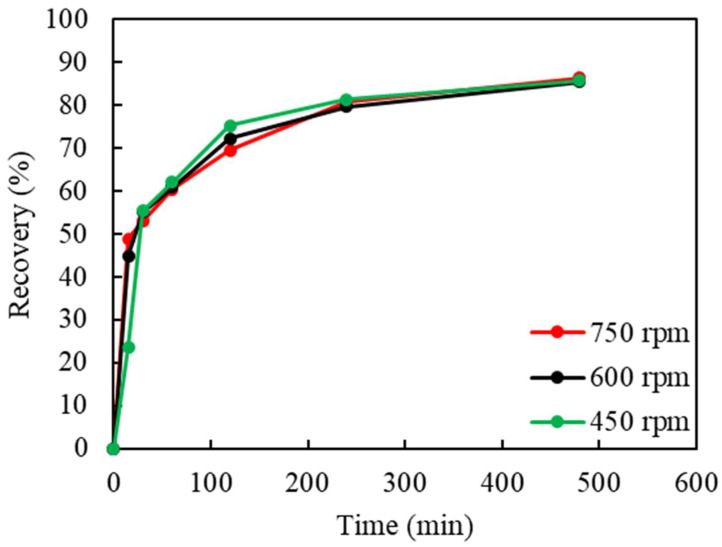
The effect of the stirring rate (rpm) on Cu recovery in ammoniacal leaching. (S/L ratio: 50 g/L, (NH_3_)_2_SO_4_: 1 M, NH_4_OH: 4 M, Cu(II): 40 g/L, particle size: −2 mm, temp. 18 °C).

**Figure 4 materials-16-06274-f004:**
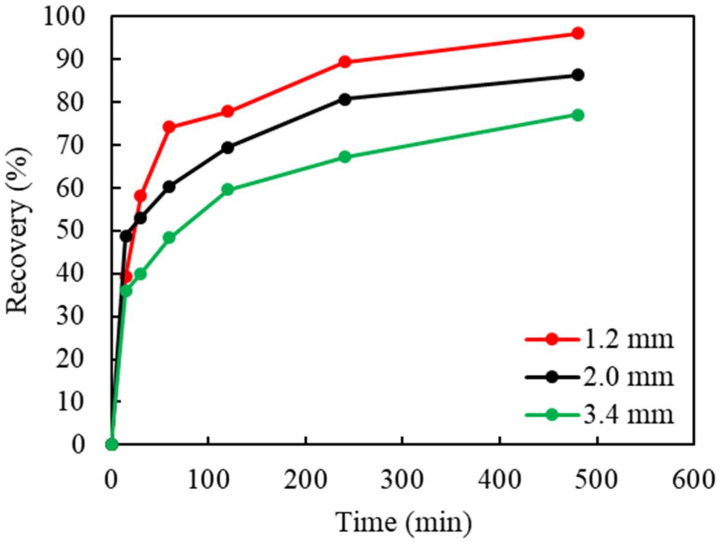
The effect of the particle size (top size, diameter, mm) on Cu recovery in ammoniacal leaching (S/L ratio: 50 g/L, (NH_3_)_2_SO_4_: 1 M, NH_4_OH: 4 M, Cu(II): 40 g/L, stirring: 750 rpm, temp.: 18 °C).

**Figure 5 materials-16-06274-f005:**
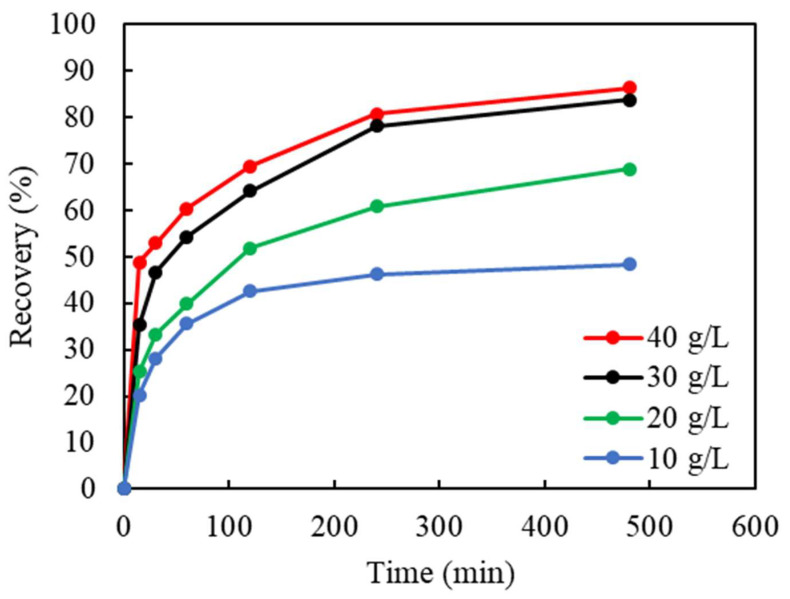
The effect of the initial Cu(II) concentration (g/L) on Cu recovery in ammoniacal leaching (S/L ratio: 50 g/L, (NH_3_)_2_SO_4_: 1 M, NH_4_OH: 4 M, particle size: −2 mm, stirring: 750 rpm, temp.: 18 °C).

**Figure 6 materials-16-06274-f006:**
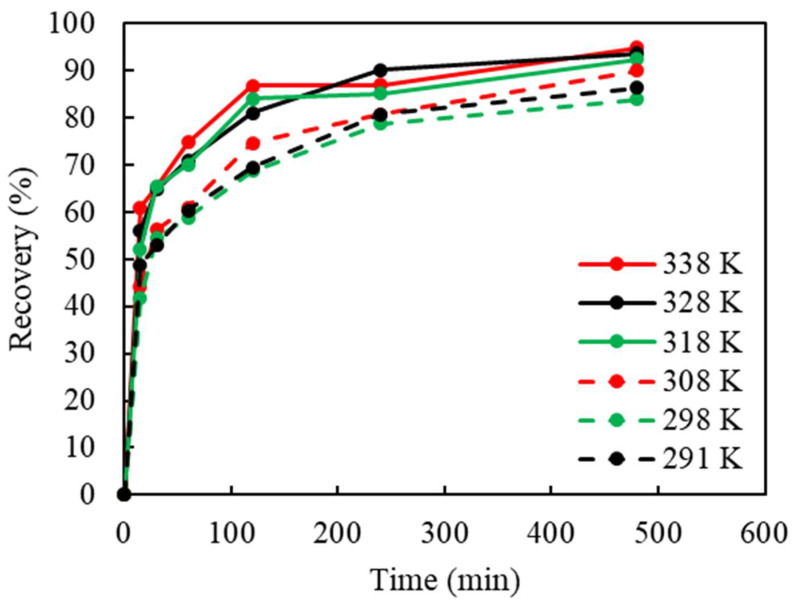
The effect of the temperature (°C) on Cu recovery in ammoniacal leaching (S/L ratio: 50 g/L, (NH_3_)_2_SO_4_: 1 M, NH_4_OH: 4 M, Cu(II): 40 g/L, particle size: −2 mm, stirring: 750 rpm).

**Figure 7 materials-16-06274-f007:**
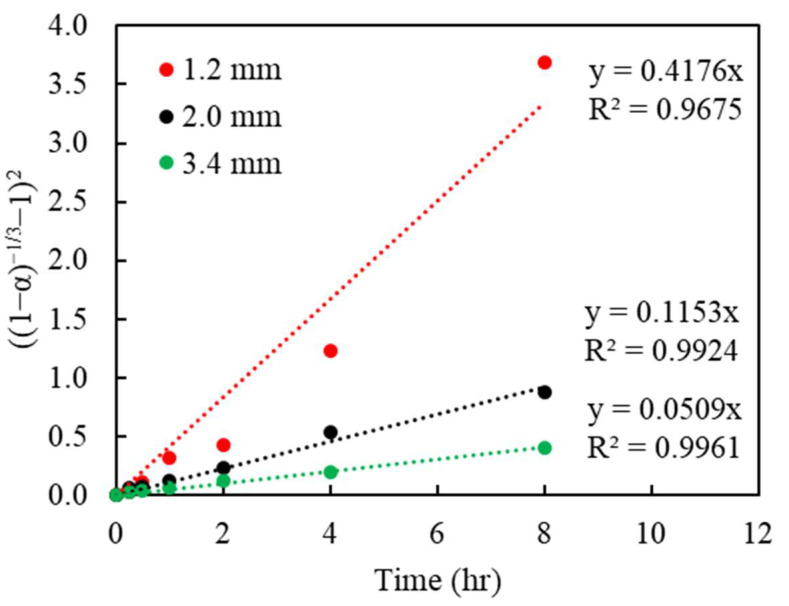
Plot of ${\left({\left(1-\alpha \right)}^{-\frac{1}{3}}-1\right)}^{2}$ vs. time under various particle sizes (diameter, mm); the data correspond to those in Figure 4.

**Figure 8 materials-16-06274-f008:**
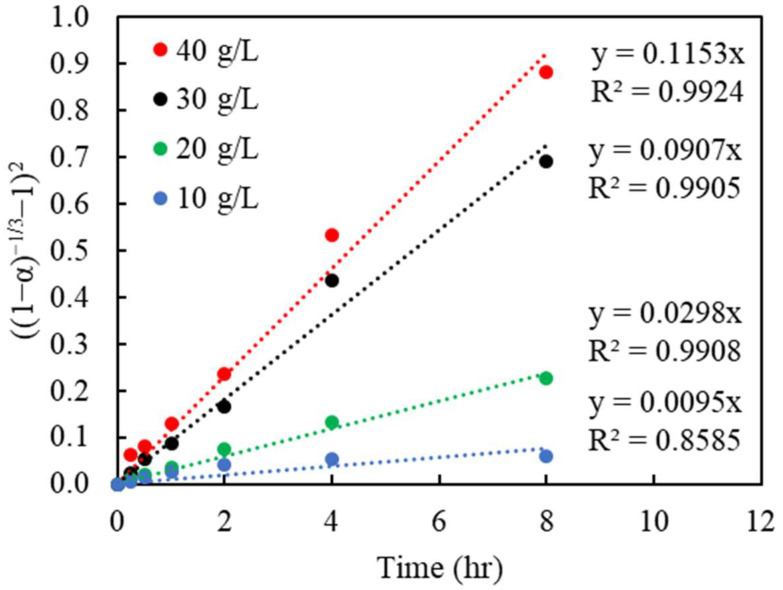
Plot of ${\left({\left(1-\alpha \right)}^{-\frac{1}{3}}-1\right)}^{2}$ vs. time under various initial Cu(II) concentrations (g/L); the data correspond to those in Figure 5.

**Figure 9 materials-16-06274-f009:**
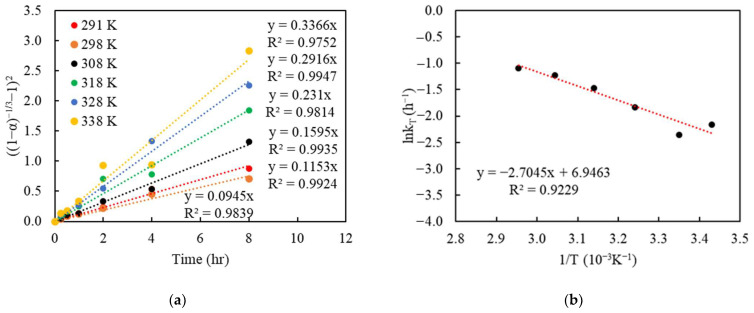
Plot of ${\left({\left(1-\alpha \right)}^{-\frac{1}{3}}-1\right)}^{2}$ vs. time under various temperatures (°C) (**a**), and the Arrhenius plot for Cu ammoniacal leaching (**b**); the data correspond to those in Figure 6.

**Figure 10 materials-16-06274-f010:**
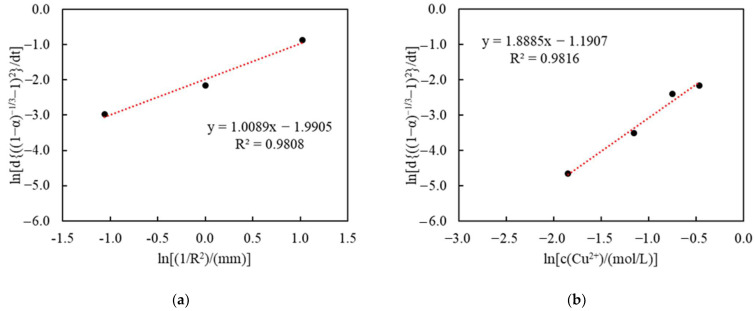
Estimation of the reaction order: plot of $ln\left[\frac{d}{dt}\left\{{\left({\left(1-\alpha \right)}^{-\frac{1}{3}}-1\right)}^{2}\right\}\right]vs.\ln\left[\left(\frac{1}{R^{2}})/(mm\right)\right]$ (**a**), and plot of $ln\left[\frac{d}{dt}\left\{{\left({\left(1-\alpha \right)}^{-\frac{1}{3}}-1\right)}^{2}\right\}\right]vs.\ln\left[c({Cu}^{2+})/(mol/L)\right]$ (**b**).

**Figure 11 materials-16-06274-f011:**
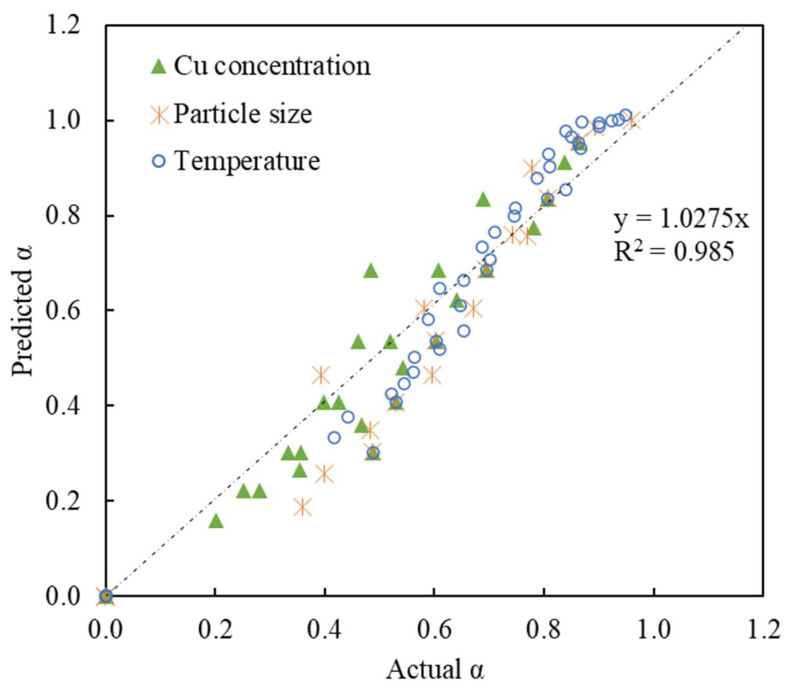
Comparison of the model-predicted reacted fraction with that of the experimental results, using Equation (15).

**Figure 12 materials-16-06274-f012:**
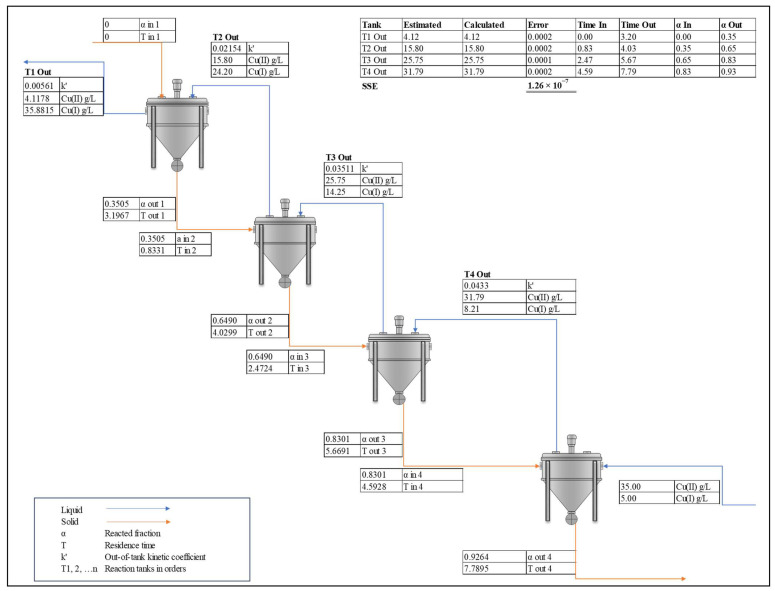
Graphic illustration of the designed CCL showing inputs, outputs, sum of square error (SSE) minimization, and the calculation of the virtual time in and out of tanks.

**Figure 13 materials-16-06274-f013:**
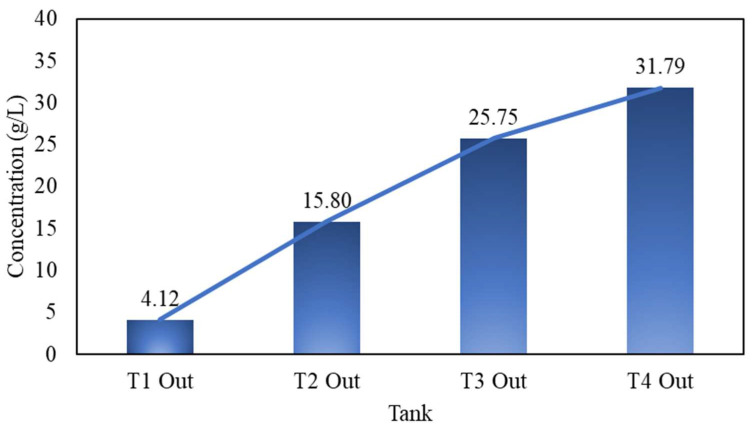
The out-of-tank Cu(II) concentration (g/L), estimated via the developed model.

**Figure 14 materials-16-06274-f014:**
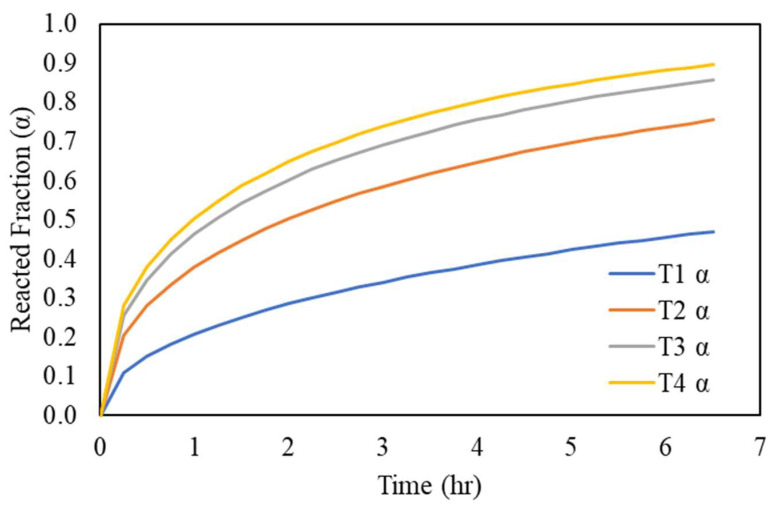
The predicted reaction fraction (α) in each leaching stage under various initial Cu(II) concentrations, according to the developed model.

**Figure 15 materials-16-06274-f015:**
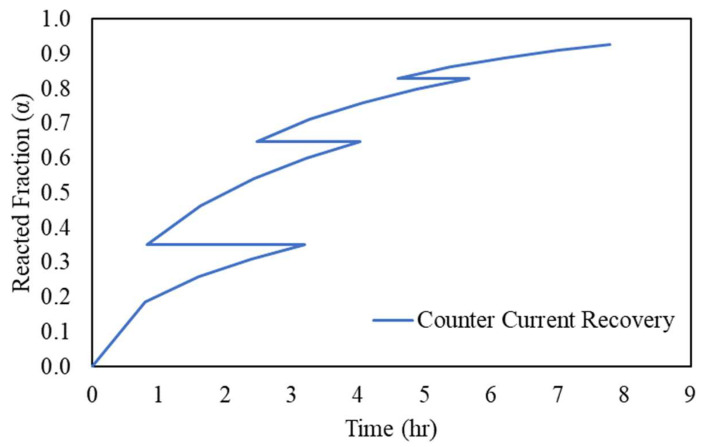
The accumulated reacted fraction (α) in the CCL circuit, predicted via the developed model.

**Table 1 materials-16-06274-t001:** A summary of the critical parameters/conditions in bench-scale Cu leaching.

Leaching Agent	Sampling Time	Particle Size	Cu(II) Conc.	S/L Ratio	Temp.	Agitation	Ar Flow Rate
mol/L	min	mm	g/L	g/L	°C	rpm	L/min
1 M (NH_4_)_2_SO_4_ and 4 M NH_4_OH	15, 30, 60, 120, 240 and 480	1.2, 2.0, 3.4	10, 20, 30, 40	50	18, 25, 35, 45, 55, 65	450, 600, 750	0.1

**Table 2 materials-16-06274-t002:** The contents of the main elements in waste RAM chips.

Al	Au	Bi	Co	Cr	Cu	Fe	Ga	Ge	Mg	Mn	Ni	Pb	Sb	Sn	Ta	Zn
ppm	ppm	ppm	ppm	ppm	%wt.	%wt.	ppm	ppm	ppm	ppm	%wt.	ppm	ppm	ppm	ppm	ppm
8629	693	308	1224	447	30.97	1.81	102	116	472	775	1.53	2584	695	6448	39	338

**Table 3 materials-16-06274-t003:** The correlation coefficient (R^2^) of Cu extraction in the Zhuravlev model under different particle sizes (d_p_) (the data correspond to those in Figure 7).

D_p_	Zhuravlev Model
mm	${\left({\left(1-\alpha \right)}^{-\frac{1}{3}}-1\right)}^{2}$
1.2	0.9675
2.0	0.9924
3.4	0.9961
Avg	0.9853

**Table 4 materials-16-06274-t004:** The correlation coefficient (R^2^) of Cu extraction in the Zhuravlev model under different initial Cu(II) concentrations (c(Cu^2+^)) (the data correspond to those in Figure 8).

c(Cu^2+^)	Zhuravlev Model
g/L	${\left({\left(1-\alpha \right)}^{-\frac{1}{3}}-1\right)}^{2}$
10	0.8585
20	0.9908
30	0.9905
40	0.9924
Avg	0.9581

**Table 5 materials-16-06274-t005:** The input parameters for the process simulation using the justified model.

Input	Symbol	Value	Unit
Rate coefficient (variable)	b	0.001363	Unitless
Feed particle size	R	1	mm
Leaching temperature	T	293	K
Initial Cu(II) concentration in solution	Cu(II)	35	g/L
Initial Cu(I) concentration in solution	Cu(I)	5	g/L
Initial Cu(0) concentration in feed	Cu(0)	30	%wt.
Lixiviant flow rate	Q_lix_	500	L/min
Mass flow of feed	Q_feed_	3.33	t/h
Mass flow of Cu(0)	Q_Cu(0)_	1	t/h

**Table 6 materials-16-06274-t006:** The calculated α in each leaching tank according to the justified model.

Tank #	Effective Leaching Time (h)	[Cu^2+^]_out_ (g/L)	Cumulative α
Tank 1	3.20	4.12	0.35
Tank 2	4.03	15.80	0.65
Tank 3	5.67	25.75	0.83
Tank 4	7.79	31.79	0.93

## Data Availability

Available upon request.

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
