# Peer review of "Kinetics and Modeling of Counter-Current Leaching of Waste Random-Access Memory Chips in a Cu-NH3-SO4 System Utilizing Cu(II) as an Oxidizer"

_materials, 2023, doi:10.3390/ma16186274_

Round 1

Reviewer 1 Report

Some comments are attached.

Author Response

As per the communication with editors, this report seems to be misplaced. Please see the attached email for the proof of waiver of this reviewer's report. 

Reviewer 2 Report

This is a nice contribution with systematic investigation by showing a thorough analysis and interpretation. The achieved results are neatly discussed. Hence, this work may be considered for publication with a minor revision.

1. The manuscript length may be reduced by moving some figures to SI.

2. Standard deviation may be provided for kinetics experiments.

see my report

Author Response

The authors are grateful for the reviewer's comments to help greatly improve the quality and readability of this manuscript. Please kindly refer to the attached file for the authors' responses and the revisions accordingly. 

Reviewer 3 Report

# Introduction

The item is too long. It could be reorganized to a brief presentation.

# Methodology, Results and Conclusions

References for Figure 2 are missing.

All analytical methods must be clearly presented in paper. Equipment must be presented by means of model, manufacturer while software must be detailer through version, manufacturer. Please check if every step from experimental procedure to chemical analysis/materials characterization and software use are covered in that way.

Figure 14 in unreadable in paper. Please make sure resolution and font size are adequate for reading and comprehension in non-digital presentation also. Consider to move Inputs, Outputs and Calculations to a separate Table.

# Overall

Adequate number of references.

Abstract is too long and must be presented in a more concise approach.

The manuscript could be an interesting reference after minor corrections.

Author Response

(The authors gave the same response as above.)

Reviewer 4 Report

In this study, an interesting original approach was chosen, some useful technical results were obtained that can be used in the future in conditions of larger-scale tests.

At the same time, I have a number of comments and questions:

1. In Fig. 8, a rather high value of R2 (96%) for 1.2 mm is questionable, given the wide spread of point values. Fig. 9 the same for 40 g/l (more than 99%?).

2. Different reaction order on page 14. Equation 16 - incorrect activation energy value? The explanations on lines 441-450 do not give a complete answer to the above questions.

3. Figure 13 is trivial, it makes no sense to give it, you can give a description of the text. In addition, it duplicates Fig. 14.

4. It is not clear why the specified ranges of parameters discussed in Section 3.2.

5 are chosen. In the conclusions, the first paragraph seems superfluous.

6. In the conclusions, the given parameters are not optimal, because they are obtained without using mathematical methods of statistical data processing.

7. It is not correct to conclude that the mass transfer process is limited only by the calculated kinetic values obtained. It was appropriate to cite here additional research results confirming the assumptions made, for example, the results of microscopic and other studies. For example, I propose to draw the attention of the authors to the following works: 

1) Hidalgo, T., et.al., 2019. Kinetics and mineralogical analysis of copper dissolution from a bornite/chalcopyrite composite sample in ferricchloride and methanesulfonic-acid solutions. Hydrometallurgy 188, 140–156.

2) Gok, O.; Anderson, et.al. Leaching kinetics of copper from chalcopyrite concentrate in nitrous-sulfuric acid. Physicochem. Probl. Mi. 2014, 50, 399–413.

3) Rogozhnikov, D.A., et.al., 2021. Kinetics and mechanism of arsenopyrite leaching in nitric acid solutions in the presence of pyrite and Fe(III) ions. Hydrometallurgy 199, 105525.

8. Has any preliminary technical and economic assessment of the proposed original method been carried out? Otherwise, the importance and necessity of this study may be questioned.

I suggest that the authors finalize the manuscript according to the above comments.

Author Response

(The authors gave the same response as above.)

Round 2

Reviewer 4 Report

I am satisfied with the changes made to the text of the manuscript and the responses to my comments. Thanks to the authors for the work done.

I recommend the manuscript for publication in this revised form.